# Cumulative Inflammation and HbA1c Levels Correlate with Increased Intima-Media Thickness in Patients with Severe Hidradenitis Suppurativa

**DOI:** 10.3390/jcm10225222

**Published:** 2021-11-09

**Authors:** Manuel Sánchez-Díaz, Luis Salvador-Rodríguez, Trinidad Montero-Vílchez, Antonio Martínez-López, Salvador Arias-Santiago, Alejandro Molina-Leyva

**Affiliations:** 1Dermatology Unit, Hospital Universitario Virgen de las Nieves, IBS Granada, 18002 Granada, Spain; manolo.94.sanchez@gmail.com (M.S.-D.); l.salvador.rodriguez1991@gmail.com (L.S.-R.); tmonterov@gmail.com (T.M.-V.); antoniomartinezlopez@aol.com (A.M.-L.); alejandromolinaleyva@gmail.com (A.M.-L.); 2Dermatology Department, School of Medicine, University of Granada, 18016 Granada, Spain; 3Hidradenitis Suppurativa Clinic, Hospital Universitario Virgen de las Nieves, 18002 Granada, Spain

**Keywords:** hidradenitis suppurativa, cardiovascular risk, intima-media thickness

## Abstract

Hidradenitis suppurativa (HS) is a chronic inflammatory skin disease that has been associated with a greater risk of metabolic and cardiovascular comorbidities. The aim of this study is to assess cardiovascular risk by means of intima-media thickness (IMT), metabolic syndrome, and other potential biomarkers in patients with severe hidradenitis suppurativa who are candidates for biologic therapy and to explore potentially associated factors. A cross-sectional study was performed. Body mass index (BMI), carotid intima-media thickness (IMT), and blood tests, including glycemic and lipid profile, insulin, vitamin D, and inflammation markers were performed. Fifty patients were included in the study; the male/female ratio was 3:2. The mean age was 38 years, and the mean disease duration was 21.8 years. The mean carotid IMT was 651.39 μm. A positive association of IMT with disease duration, tobacco consumption, and HbA1c levels was observed. HbA1c correlated with the age of onset, hypertension, metabolic syndrome, and glucose levels. Vitamin D levels inversely correlated with the number of areas affected. In conclusion, patients with severe HS present a higher cardiovascular risk, but it is not distributed equally within the patients: Tobacco consumption, inadequate glycemic control, and disease duration could be useful clinical and biochemical markers to identify patients at higher risk.

## 1. Introduction

Cardiovascular diseases are a leading global cause of morbidity and mortality [1]. Different clinical risk factors have been related to an increase in the cardiovascular risk in patients: aging [2], obesity [3], specific dietary patterns [4], or lack of physical activity [5] are some of the most important. Moreover, several blood markers have been related directly or indirectly, acting through the increase in the low-grade chronic pro-inflammatory state to an increase in cardiovascular risk: high levels of glucose, glycated hemoglobin (HbA1c) and diabetes mellitus [6], high levels of total cholesterol and specifically low-density-lipoprotein (LDL) cholesterol [7], low levels of vitamin D [8], or high levels or C-reactive protein (CRP) [9] are some examples. Atherosclerosis, which is currently accepted as the major direct cause of cardiovascular disease, is related to all these factors, and it is caused by low-grade chronic inflammation [10,11]. Moreover, metabolic syndrome, which is also associated to a higher cardiovascular risk, has been previously associated with HS [12].

On the one hand, several studies address the issue that poor glycemic control is associated with HS [13,14]. Moreover, it is widely known that glycemic disorders are strongly associated with cardiovascular risk [15]. On the other hand, vitamin D deficiency has previously been associated with a higher inflammation burden in HS [16]. Furthermore, it seems that the deficiency of vitamin D can be related to the pathogenesis of the disease [17], and, although controversial, it seems that vitamin D deficiency could be related to cardiovascular events in some specific groups [18,19], including psoriasis patients [20].

Recently, different chronic inflammatory diseases have been associated themselves with a higher cardiovascular risk: systemic lupus erythematosus [21], inflammatory bowel disease [22], rheumatoid arthritis [23], psoriasis [24,25,26], or hidradenitis suppurativa [27,28] are among them.

Hidradenitis suppurativa is a chronic inflammatory condition primarily affecting apocrine-gland-rich regions of the body such as the axillary and groin areas, presenting with painful nodules and abscesses that may coalesce and form fistulas where the pus may drain [29].

Given the important implications of the factors associated with cardiovascular risk and the high burden of inflammation that is associated to severe hidradenitis suppurativa, the aim of this study is to assess cardiovascular risk by means of intima media thickness (IMT) and analyze other proatherosclerotic biomarkers in patients with severe hidradenitis suppurativa who are candidates to biologic therapy and to explore potentially associated clinical and biochemical factors.

## 2. Materials and Methods

Design: A cross-sectional study was performed to assess IMT in patients suffering from severe HS who are candidates to biologic treatments, and to evaluate potential clinical and biomarkers factors associated.

Patients: Patients included in the study are receiving health care in the Hidradenitis Suppurativa Clinic of the Virgen de las Nieves University Hospital. The patients are part of a prospective cohort in which socio-demographic, clinical, and analytic data are collected. This study was approved by the Institutional Review Board of the Hospital Universitario Virgen de las Nieves and is in accordance with the Declaration of Helsinki.

Inclusion criteria: The inclusion criteria were as follows: (a) Patients with clinical diagnosis of HS; (b) Patients suffering from severe HS who are candidates to be treated with biologic drugs [30]; (c) Informed consent from the patient of the legal representatives to be included in the study.

Exclusion criteria: The exclusion criteria were as follows: (a) Refusal from the patient or legal representative to participate in the study; (b) Patients suffering from HS who are not candidates to be treated with biologic drugs.

### 2.1. Variables of Interest

#### 2.1.1. Main Variables

The patients were assessed the day of the initiation of their biologic treatment before the first dose was administered. Main variables include variables related to the disease severity and variables related to cardiovascular risk assessment.

Disease severity was assessed using disease activity indexes:

The International HS severity scoring system (IHS4) was used to assess inflammatory activity. It was calculated using the following formula: (number of nodules × 1) + (number of abscesses × 2) + (number of fistulas × 4) [31].The Hurley classification was used to assess structural damage. It consists of 3 stages (I, presence of abscesses without fistulous tracts or scars; II, recurrent abscesses and single or multiple fistulae and scars widely separated between them; III, abscesses and confluent fistulas with large areas of extensive scarring) [32].Number of body areas affected by HS lesions.To assess symptoms, the Numeric Rating System (NRS) was used for pain, itching and suppuration. Values range between 0 and 10 [33].

Cardiovascular risk factors were evaluated by performing blood tests, clinical examination, and carotid artery ultrasonography:

Carotid intima-media thickness (IMT): IMT was assessed in both carotid arteries of the patients by using ultrasonography. Left, right, and mean IMT was recorded. The procedure was performed as previously described in similar investigations [25]: A complete examination of the left and right common carotid arteries was performed. Patients were in supine position with their necks being 60° laterally inclined. Six IMT measurements were performed, and in all of them, the probe was 10 mm away from the carotid bifurcation. Only those measurements with a standard deviation inferior to 20 μm were taken as valid measures. A high-resolution ultrasound scanner (Esaote MyLab Gold 25) with a 12 mHz sounding line running specific software for measuring ITM (Esaote QIMT Software) was employed. The definitive value of the IMT measurements was obtained by calculating the average of 6 valid measurements taken from the sonographic scanning. Figure 1 shows the IMT assessment.Blood biomarkers: Different measurements were performed in patients following an 8 h fasting period. Recorded blood parameters included: basal glucose, basal insulin, HOMA score, glycated hemoglobin, triglycerides, HDL and LDL cholesterol, transaminases, albumin, parathyrin, vitamin D, erythrocyte sedimentation rate (ESR), and C-Reactive Protein (CRP).Clinical data: Patients were explored to assess the Body Mass Index (BMI), the metabolic syndrome criteria (ATPIII criteria) [34], and tobacco consumption (cigarettes/day).

#### 2.1.2. Other Variables

Socio-demographic, biometric and clinical variables, including age, sex, comorbidities, and previous treatments for HS, were recorded by clinical interview, physical examination, and cutaneous ultrasonography through the use of a 7–15 MHz linear probe (myLab7 Esaote, Genoa, Italy).

### 2.2. Statistical Analysis

Descriptive statistics were used to evaluate the characteristics of the sample. The Shapiro–Wilk test was used to assess the normality of the variables. Continuous variables are expressed as mean and standard deviation (SD). Qualitative variables are expressed as relative and absolute frequency distributions. The χ2 test or Fisher’s exact test, as appropriate, were used to compare nominal variables, and Student’s *t*-test or Wilcoxon–Mann–Whitney test were used to compare between nominal and continuous data. To explore possible associated factors, simple linear regression was used for continuous variables. The β coefficient and SD were used to predict the log odds of the dependent variable. Significantly associated variables (*p* < 0.05) or those showing trends toward statistical significance (*p* < 0.20) were included in multivariate analysis. Multivariate logistic regression analyses were carried out to identify the factors associated with target variables. Statistical significance was considered if p values were less than 0.05. Statistical analyses were performed using JMP version 9.0.1 (SAS institute, California, NC, USA).

## 3. Results

### 3.1. Socio-Demographic and Clinical Features of the Sample

Fifty patients with severe HS were included in the study. The male/female ratio was 3:2, and the mean age of the sample was 38 years old (SD 12.9). Family history of HS was present in 44% (22/50) of the patients; see Table 1. Regarding HS severity, the mean number of areas affected indicated was 4.53 (SD 3.10). The IHS4 mean score was 21.6 (SD 12.59), which indicated severe disease. Furthermore, 56% (28/50) of the patients were classified as Hurley III stage. Patients had suffered from HS for a mean of 16.6 years (SD 13.54). The subjective severity index NRS for pain, pruritus, malodor, suppuration, and general NRS showed high scores, with means in all cases above 5 points; see Table 1. Acne conglobate and pilonidal sinus were the most common comorbidities in the sample (36% (18/50) and 38% (19/50), respectively). IHS4 was found to be positively correlated with disease duration, BMI, number of areas affected, Hurley stage, ESR, and CRP (*p* < 0.05). Moreover, IHS4 was inversely correlated with vitamin D levels (*p* = 0.04).

### 3.2. Distribution of Cardiovascular Risk Markers in the Sample

Previously diagnosed prevalence of diabetes mellitus and dyslipidemia was 10% (5/50) for both entities; see Table 2. The basal glycemia, basal insulin, and mean HOMA index indicated a high prevalence of insulin resistance in the sample: the mean HOMA index was above the threshold of 3, which is considered to diagnose insulin resistance. Obesity had a high prevalence in the sample and appeared in 42% (21/50) of the patients with a mean BMI of 30.36 (SD 7.45). The mean abdominal circumference was 98.18 cm (SD 16.26). High blood pressure had been diagnosed in 20% (10/50) of the patients, and the mean tobacco consumption was 4.84 cigarettes/day (SD 8.15). Mean vitamin D values were below the threshold of 20 mg/dL, indicating vitamin D deficiency; see Table 2. Finally, 26% (13/50) of the patients met the criteria for metabolic syndrome.

### 3.3. Glycemic Disorders

Univariate analyses showed that increasing body mass index, metabolic syndrome criteria, and basal fasting glucose correlated with HbA1c levels. Moreover, high blood pressure and the age of onset showed trends toward statistical significance. No other factors were found to be related to HbA1c in univariant analyses. After multivariate analysis, age of onset, high blood pressure, metabolic syndrome, and basal glucose were correlated to HbA1c levels; see Table 3.

### 3.4. Vitamin D

Vitamin D levels were inversely associated with the number of areas affected, IHS4 score, and Hurley stage in univariant analysis. No association was found with parathyrin levels. The multivariate analysis showed an inverse correlation between the number of areas affected and the vitamin D levels; see Table 3.

### 3.5. Intima-Media Thickness and Potential Associated Factors

Mean IMT was 651.30 μm (SD 130.9). The potential predictors of IMT were explored in univariate analyses; see Table 2. Age, age of onset, disease duration, and glycated hemoglobin levels (HbA1c) were found to be positively correlated to IMT. In addition, basal insulin level was inversely correlated to IMT. Tobacco consumption and HOMA index showed trends toward statistical significance. After multivariate analysis, a positive association with disease duration, tobacco consumption, and HbA1c levels was observed, whereas no association was found with insulin or HOMA index; see Table 4.

## 4. Discussion

The results of our study show that the IMT is associated with cumulative inflammation and long-term glycemic control in patients with severe HS. We have also observed that HbA1c levels are linked to the presence of metabolic syndrome and that vitamin D levels inversely correlate with disease severity and extension. Patients with severe HS with active tobacco consumption, long disease duration, and metabolic comorbidities, especially abnormal glucose values or diabetes mellitus, may represent a subset of patients with particularly elevated cardiovascular risk.

IMT is an useful tool to assess cardiovascular risk [35] as a marker of subclinical atherosclerosis. This method has been widely used to predict cardiovascular events [36] and to assess the prognosis of patients suffering from cardiovascular diseases [37]. Moreover, it is useful to assess cardiovascular risk in inflammatory diseases, such as rheumatoid arthritis [38], systemic lupus erythematosus [39], inflammatory bowel disease [40], or psoriasis [25]. On the other hand, tobacco consumption [41] and glycemic disorders [42] are known cardiovascular risk factors that have shown to be correlated to IMT in previous studies. This is in line with the findings of the present study, as these factors correlate to IMT in patients with HS.

On the other hand, the disease duration, as a marker of the cumulative burden of inflammation in HS, has correlated to IMT in our study. Previous reports [25,43,44] agree on the importance of the inflammatory burden, which is measured as high activity indexes or disease duration and their relationship with increased IMT for patients with rheumatologic disorders and psoriasis. Finally, although some studies address the issue of IMT in HS [45], to the best of our knowledge, there are no other specific reports assessing IMT-related factors in HS patients.

Despite its strengths, IMT might be inaccurate in some cases, given its operator-dependent nature [46], although this weakness can be overcome by the use of specific software, such as the one used in the present study, to assist in the measurement process. Moreover, the process of imaging and obtaining the thickness can be time consuming, and there is a lack of standardized normal values, especially in patients with no other cardiovascular risk factors [47]. In this regard, the combination of IMT with other techniques may be of interest. Pulse wave velocity (PWV) has emerged as a measure of arterial stiffness, which also correlates to cardiovascular risk in different subsets of patients and diseases [48,49,50,51]. The addition of this technique to cardiovascular risk studies might improve its accuracy and provide new a novel insight into associated factors. Moreover, this technique would be easier to apply in the outpatient setting, enabling cardiovascular risk assessment without the need for an ultrasound scanner and with less time consumption.

Diabetes mellitus and, specifically, high HbA1c levels have been linked to an increase in the cardiovascular risk [52,53]. In addition, insulin resistance and diabetes mellitus are more prevalent in patients with HS when compared to controls [13,54]. The present study shows that HbA1c levels in patients with HS correlates to IMT, which is consistent with previous reports, but this had not yet been clearly established in particular in patients with HS. Metformin, an anti-diabetic drug that also has anti-inflammatory properties [55], might play a specific role in these patients by both improving the glycemic levels (and therefore decreasing their cardiovascular risk) and because of its proper anti-inflammatory effect, whose efficacy has already been reported in patients with HS [56].

Although still controversial, several studies have associated the low levels of vitamin D with cardiovascular risk [8,18,19]. Moreover, vitamin D is thought to be related to chronic inflammation because of its potential anti-inflammatory activity [57,58], and it has been related to higher inflammatory indexes in some diseases, including HS [59,60]. These facts were in line with the findings of our study: a higher number of areas affected, which indirectly indicate the inflammatory burden of the patients, were associated with lower levels of vitamin D. In addition, lower levels of vitamin D were associated with a higher IHS4 index. Whether low vitamin D is a cause or a consequence of the high inflammatory burden still remains unknown.

On the other hand, as it has been recently described, HS, as a chronic inflammatory disease, is related to iron deficiency [61]. Since iron deficiency has been associated with higher cardiovascular risk [62], future studies addressing the relationship between iron deficiency and cardiovascular risk in HS would be of great interest. In this line, it would be also of importance to clarify the role of circulating endothelial progenitor cells in cardiovascular risk in patients with HS [63].

Given that most biologic drugs used for HS act through the modulation of cytokines, their powerful anti-inflammatory effect could be related to the improvement of cardiovascular risk. This fact has been reported for other disorders where more studies have been performed. For example, it is known that biologic therapy in psoriasis patients leads to a decrease in the cardiovascular risk measured by the atherosclerosis plaque burden [64] or the decrease in the number of cardiovascular events [65]. Other similar reports have been published for chronic inflammatory diseases [66]. This relationship seems to be clear for anti-TNF and anti-IL6 agents, and it remains still uncertain for other biologic drugs [67]. In addition, recent reports include the improvement of insulin resistance and that pancreatic beta cells have beneficial effects on the anti-TNF therapy in rheumatoid arthritis [68], which supports its potential for improvement cardiovascular risk.

Future studies based on the follow-up of the present cohort might be useful to assess whether biologic therapies are capable of modulating the cardiovascular risk and glycemic parameters of patients suffering from severe HS. Moreover, the differential analysis including different biologic therapies and targets might help dermatologists choose the most appropriate drug for patients with HS and significant cardiovascular risk.

The main limitation of the present study is its cross-sectional nature, which makes it impossible to assess causality. Moreover, a higher number of patients could have been useful to find potential correlations between variables that have not been shown in the present analysis.

## 5. Conclusions

Severe HS involves a great inflammatory burden and a higher cardiovascular risk. Cumulative inflammation and glycemic disorders could be useful to identify patients with significant cardiovascular risk and are potential targets to be improved in the treatment of this disease.

## Figures and Tables

**Figure 1 jcm-10-05222-f001:**
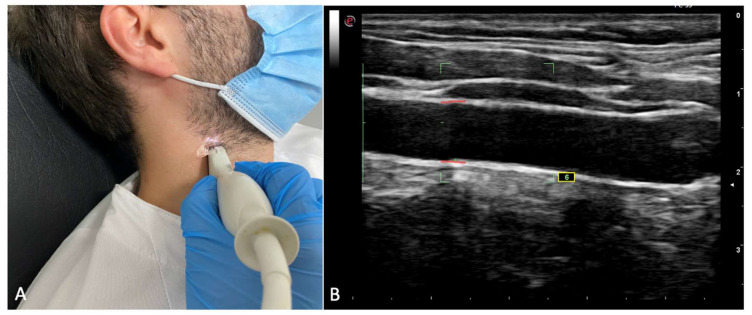
Figure showing ultrasonographic intima-media thickness (IMT) assessment. (**A**) Position of the patient and US probe. (**B**) Image showing common carotid artery. Once the common carotid artery is detected, IMT is measured.

**Table 1 jcm-10-05222-t001:** Socio-demographic and clinical features of the sample.

Variables All Patients (*n* = 50)			
Age (years)	38 (SD 18.85)	Pilonidal sinus (%)	38% (19/50)
Age of onset (years)	21.83 (SD 9.81)	Number or areas affected	4.53 (SD 2.06)
Disease duration (years)	16.6 (SD 13.54)	NRS pain	6.84 (SD 3.10)
Family history of HS (%)	44% (22/50)	NRS pruritus	5.84 (SD 3.23)
Previous biologic treatment (%)	22% (11/50)	NRS malodor	6.36 (SD 3.31)
Previous surgical interventions for HS (%)	34% (17/50)	NRS suppuration	6.62 (SD 3.02)
Dissecting cellulitis of the scalp (%)	10% (5/50)	General NRS	6.68 (SD 2.69)
Acne conglobate (%)	36% (18/50)	IHS4 index	21.6 (SD 12.59)
Hurley Stage (%)	I: 8% (4/50)	II: 36% (18/50)	III: 56% (28/50)

**Table 2 jcm-10-05222-t002:** Cardiovascular risk markers in the sample.

Cardiovascular Risk Markers All Patients (*n* = 50)			
Diabetes mellitus (%)	10% (5/50)	Dyslipidemia (%)	10% (5/50)
Basal glycemia (mg/dL)	90.5 (SD 15.71)	Triglycerides (mg/dL)	130.13 (SD 62.72)
Basal insulin (microU/mL)	15.17 (SD 23.43)	HDL (mg/dL)	51.71 (SD 17.87)
HOMA index	3.79 (SD 7.74)	High blood pressure (%)	18% (9/50)
HbA1c (%)	5.57 (SD 0.73)	Systolic blood pressure (mmHg)	129.08 (SD 7.75)
Tobacco (cig/day)	4.84 (SD 8.15)	Diastolic blood pressure (mmHg)	83.44 (SD 6.81)
Obesity (%)	42% (21/50)	Metabolic syndrome (%)	26% (13/50)
Body Mass Index (kg/m^2^)	30.36 (SD 7.45)	Vitamin D (ng/mL)	19.80 (SD 7.08)
Abdominal circumference (cm)	98.18 (SD 16.26)	Intima-media thickness (μm)	651.30 (SD 130.90)

**Table 3 jcm-10-05222-t003:** Potential clinical and socio-demographic features associated to HbA1c and vitamin D levels.

Factors	HbA1c				Vitamin D			
	Univariate Analysis		Multivariate Analysis		Univariate Analysis		Multivariate Analysis	
	Difference/β	*p* Value	β	*p* Value	Difference/β	*p* Value	β	*p* Value
Sex	Male: 5.49 (SE 0.09)	0.95	-	-	Male: 19.76 (SE 1.34)	0.99	-	-
	Female: 5.49 (SE 0.12)				Female: 19.78 (SE 1.62)			
Age	0.006 (SE 0.008)	0.35	-	-	−0.13 (SE 0.08)	0.09	−0.08 (SE 0.07)	0.26
Age of onset	0.01(SE 0.007)	0.16	0.011 (SE 0.006)	0.046	0.047 (SE 0.11)	0.66	-	-
Duration of disease	−0.002 (SE 0.005)	0.75	-	-	−0.13 (SE 0.07)	0.09	-	-
Family history	Yes: 5.52 (SE 0.11)	0.65	-	-	Yes: 20.48 (SE 1.39)	0.54	-	-
	No: 5.47 (SE 0.10)				No: 19.19 (SE 1.39)			
Previous biologic treatments	Yes: 5.33 (SE 0.16)	0.24	-	-	Yes: 18.39 (SE 2.28)	0.28	-	-
	No: 5.54 (SE 0.09)				No: 20.12 (Se 1.15)			
Previous surgical interventions for HS	Yes: 5.43 (SE 0.13)	0.54	-	-	Yes: 19.13 (SE 1.75)	0.68	-	-
	No: 5.53 (SE 0.09)				No: 20.11 (SE 1.27)			
High blood pressure	Yes: 5.18 (SE 0.18)	0.053	0.21 (SE 0.07)	0.006	Yes: 18.97 (SE 2.40)	0.42	-	-
	No: 5.57 (SE 0.08)				No: 19.98 (SE 1.15)			
Tobacco	0.01 (SE 0.009)	0.29	-	-	0.10 (SE 0.16)	0.50	-	-
Body mass index	0.02 (SE 0.009)	0.04	0.007 (SE 0.008)	0.42	−0.042 (SE 0.14)	0.76	-	-
Metabolic syndrome	Yes: 5.87 (SE 0.13)	0.002	0.15 (SE 0.07)	0.03	1.72	0.43	-	-
	No: 5.36 (SE 0.08)							
Number of areas affected	0.008 (SE 0.04)	0.82	-	-	−1.57 (SE 0.49)	0.002	−1.17 (SE 0.57)	0.047
IHS4 index	0.007 (SE 0.006)	0.27	-	-	−0.24 (SE 0.09)	0.014	−0.0005 (SE 0.12)	0.99
Hurley	I: 5.31 (SE 0.27)	0.75	-	-	I: 26.81 (SE 3.28)	0.067	2.95 (SE 2.34)	0.21
	II: 5.48 (SE 0.3)				II: 22.12 (SE1.55)			
	III: 5.53 (SE 0.1)				III: 17.16 (SE 1.26)		0.60 (SE 1.52)	0.69
Basal glucose	0.02 (SE 0.003)	<0.001	0.017 (SE 0.004)	<0.0001	−0.08 (SE 0.06)	0.25	-	-
Basal Insulin	0.0005 (SE 0.003)	0.88	-	-	−0.027 (SE 0.04)	0.54	-	-
HOMA index	0.002 (SE 0.009)	0.82	-	-	−0.088 (SE 0.13)	0.50	-	-
HbA1c	1	<0.001	-	-	−1.20 (SE 1.40)	0.39	-	-
Triglycerides	0.002 (SE 0.001)	0.068	-	-	0.012 (SE 0.01)	0.47	-	-
LDL cholesterol	0.003 (SE 0.002)	0.12	-	-	−0.009 (SE 0.03)	0.74	-	-
Parathyrin	−0.002 (SE 0.003)	0.57	-	-	−0.056 (SE 0.04)	0.17	-	-
Vitamin D	−0.003 (SE 0.01)	0.79	-	-	1	<0.001	-	-
ESR	−0.009 (SE 0.004)	0.83	-	-	−0.042 (SE 0.05)	0.44	-	-
CRP	−0.00002 (SE 0.003)	0.99	-	-	−0.016 (SE 0.04)	0.73	-	-
Ferritin	0.001 (SE 0.009)	0.22	-	-	−0.0005 (SE 0.01)	0.97	-	-
R^2^	-		0.5848		-		0.2968	

**Table 4 jcm-10-05222-t004:** Potential clinical and socio-demographic features associated to intima-media thickness.

Factors	Univariate Analysis		Multivariate Analysis	
	Difference/β	*p* Value	β Coefficient	*p* Value
Sex	Male: 659 (SD 23.9)	0.77	-	-
	Female 648 (SD 30)			
Age	5.64 (SE 1.21)	<0.001	-	-
Age of onset	4.96 (SE 1.80)	<0.01	-	-
Duration of disease	3.53 (SE 1.30)	<0.01	3.61 (SE 1.51)	0.02
Family history	Yes: 626.57 (SD 27.34)	0.17	-	-
	No: 678.37 (SD 24.68)			
Previous biologic treatments	Yes: 663.59 (SD 39.45)	0.84	-	-
	No: 652.66 (SD 21.22)			
Previous surgical interventions for HS	Yes: 673.29 (SD 31.58)	0.48	-	-
	No: 645.45 (SD 32.02)			
High blood pressure	Yes: 699.06 (SD 40.93)	0.17	-	-
	No: 637.60 (SD 19.67)			
Tobacco	4.86 (SE 2.62)	0.07	5.52 (SE 2.49)	0.03
Body mass index	2.12 (SE 2.50)	0.40	-	-
Metabolic syndrome	Yes: 659.85 (SD 36.3)	0.89	-	-
	No: 653.40 (SD 21.8)			
Number of areas affected	4.08 (SE 9.37)	0.67	-	-
IHS4 index	0.36 (SE 1.49)	0.81	-	-
Hurley	I: 606.87 (SE 65.29)	0.53	-	-
	II: 678.49 (SE 30.78)			
	III: 646.53 (SE 25.13)			
Basal glucose	1.14 (SE 1.19)	0.34	-	-
Basal insulin	−1.58 (SE 0.76)	0.04	−0.70 (SE 6.92)	0.92
HOMA index	−4.20 (SE 2.32)	0.07	−0.41 (SE 20.51)	0.98
HbA1c	62.60 (SE 23.9)	0.01	64.76 (SE 24.62)	0.01
Triglycerides	0.14 (SE 0.30)	0.63	-	-
LDL cholesterol	−0.88 (SE 0.53)	0.10	-	-
Parathyrin	0.95 (SE 0.73)	0.20	-	-
Vitamin D	−2.59 (SE 2.63)	0.33	-	-
ESR	−0.91 (SE 0.98)	0.20	-	-
CRP	−1.33 (SE 0.80)	0.10	-	-
Ferritin	0.39 (SE 0.24)	0.11	-	-
R^2^	-	-	0.4230	

## Data Availability

Data are contained within the article.

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
