# Peer review of "Cumulative Inflammation and HbA1c Levels Correlate with Increased Intima-Media Thickness in Patients with Severe Hidradenitis Suppurativa"

_jcm, 2021, doi:10.3390/jcm10225222_

Round 1

Reviewer 1 Report

Dear Authors.

Thank you very much for the possibility to read such an interesting manuscript 

I have only couple of minor issues:

  1. Did you have an approval of Ethics comitee? I couldn't find it in the manuscript. If yes please provide the number. I believe it is crucial.
  2. Please discuss the iron deficiency in HS and its possible impact on cardiovascular risk
  3. See if circulating endothelial progenitor cells may be of use in your manuscript, because as it was published before the level is much lower than in healthy patients

Author Response

Dear Reviewer,

Thank you very much for your comments, as they allow us to improve the quality of our work. All the suggested changes have been implemented. Here there is a point-by-point response to your comments:

  • The approval of the Ethics committee has been included in the text. This has not been previously included because of an error during manuscript writing.
  • A brief discussion about iron deficiency and cardiovascular risk has been added to the manuscript.
  • A comment regarding the issue of endothelial circulating progenitor cells has been added.

Reviewer 2 Report

Dear Authors,

I have read your article with great interest.

It is well known that diabetes, obesity and smoking are common in HS population. Therefore, adequate assessment of cardiovascular risk is crucial.

I have only one question:

If the patient comes to dermatology outpatient office for treatment of HS it is almost impossible to measure intima media using ultrasound. How would you manage this situation? I believe it will be in the majority of outpatient offices.

Author Response

Dear Reviewer,

Thank you very much for your comments, as they allow us to improve the quality of our work. Here there is a point-by-point response to your comment:

  • Given that ultrasound is widely available in dermatology consultations in our region, the assessment of IMT with carotid US would not represent a major problem.
  • However, a little comment has been added about this issue: The measure of pulse-wave velocity can be performed easily within minutes in the dermatology office, with a device similar to digital blood pressure monitors. This could be one possible solution to the problem.

Reviewer 3 Report

Problem of inflammation and hidradenitis suppurativa is an extremely important topic requiring close attention and also interdisciplinary one. Life style changes are of great importance due to Homo sapiens being one entity. Approach to the disease should be multidisciplinary and the Authors are going in this direction. Congratulations! Just continue 

Author Response

Dear Reviewer, 

Thank you very much for your comments. We are pleased to read such good comments which encourage us to continue working on this issue.

Reviewer 4 Report

An interesting original study showing that intima media thickness is associated with cumulative inflammation and long-term glycemic control in patients with severe hidradenitis suppurativa. Although the main limitations of the study are the relatively low number of patients (due to the rarity of the condition) and that a similar study was already present in literature, I think the paper will be considered after revisions:

page 2 line 54 a small description of HS is necessary;you should add: "Hidradenitis suppurativa is a chronic inflammatory condition primarily affecting apocrine-gland-rich regions of the body such as the axillary and groin areas, presenting with painful nodules and abscesses that may coalesce and form fistulas where the pus may drain. " and you should cite an article such as: doi: 10.3390/ijms21228436.

A lot of comparisons were performed...Did you use a correction for multiple comparisons? If not, I suggest you use one.

Thank You

Author Response

Dear Reviewer,

Thank you very much for your comments, as they allow us to improve the quality of our work. All the suggested changes have been implemented. Here there is a point-by-point response to your comments:

  1. The description of HS and the mentioned article have been added to the manuscript.
  2. Although multiple comparisons were performed, the statistic error associated with multiple comparisons has been controlled by the use of multivariate analysis, so as to make the multiple comparisons correction factor not necessary.

Round 2

Reviewer 4 Report

The paper is in my opinion publishable